# Stacked binding of a PET ligand to Alzheimer's tau paired helical filaments

Gregory E. Merz [1,2], Matthew J. Chalkley[3], Sophia K. Tan[3], Eric Tse [1], Joanne Lee[1], Stanley B. Prusiner [1,2,4], Nick A. Paras [1,2], William F. DeGrado[1,3] & Daniel R. Southworth [1,4] ✉

Accumulation of filamentous aggregates of tau protein in the brain is a pathological hallmark of Alzheimer's disease (AD) and many other neurodegenerative tauopathies. The filaments adopt disease-specific cross-β amyloid conformations that self-propagate and are implicated in neuronal loss. Development of molecular diagnostics and therapeutics is of critical importance. However, mechanisms of small molecule binding to the amyloid core is poorly understood. We used cryo–electron microscopy to determine a 2.7 Å structure of AD patient-derived tau paired-helical filaments bound to the PET ligand GTP-1. The compound is bound stoichiometrically at a single site along an exposed cleft of each protofilament in a stacked arrangement matching the fibril symmetry. Multiscale modeling reveals pi-pi aromatic interactions that pair favorably with the small molecule–protein contacts, supporting high specificity and affinity for the AD tau conformation. This binding mode offers critical insight into designing compounds to target different amyloid folds found across neurodegenerative diseases.

The abnormal accumulation of misfolded tau proteins in the brain occurs in a large (>25)[1] subset of neurodegenerative diseases (NDs) known as tauopathies[2,3], the most common and widely studied of which is Alzheimer's disease (AD)[4]. The spread of tau deposits, known as neurofibrillary tangles (NFTs) in AD, parallels neuronal loss and cognitive impairment[5,6] and serves as a marker for disease progression[7]. Moreover, accumulation and maturation of NFTs appear to be the final stages of a process in which soluble tau misfolds into amyloid filaments that self-propagate and transmit as prions across neurons via synaptic junctions[8]. Prions were first identified in the scrapie prion protein (PrP$^{Sc}$), which causes Creutzfeldt-Jakob (CJD), Gerstmann–Sträussler–Scheinker (GSS), and other incurable diseases[9,10] in which amyloids also accumulate with disease progression. Cryo–electron microscopy (Cryo-EM) of tau filaments purified from postmortem patient brains with different NDs has revealed that the amyloid core adopts different cross-β sheet structural conformations comprised of the tau microtubule-binding repeat[11–16]. For example, AD filaments are comprised of 3 R and 4 R isoforms and adopt a C-shaped fold[11,13], while in Pick's disease, 3 R tau forms an elongated J-shaped fold[12], and in corticobasal degeneration (CBD), 4 R tau adopts a four-layered β-strand arrangement[15]. These distinct structural conformations have opened up the possibility of binding small molecules to different tau filaments for disease-specific targeting; here, we determined a cryo-EM structure of a small molecule bound to tau that reveals a potential mechanism for achieving site-specificity.

Small molecules that can discriminate among amyloid filaments[17,18], and even strains of the same prions[19,20], have been developed. However, the basis of this specificity is unknown. Despite this limitation, a number of promising tau-selective positron-emission tomography (PET) ligands for AD have been developed and tested in vivo[21]. Many such molecules contain heterocyclic aromatic moieties,

[1]Institute for Neurodegenerative Diseases, University of California San Francisco, San Francisco, CA, USA. [2]Department of Neurology, University of California San Francisco, San Francisco, CA, USA. [3]Department of Pharmaceutical Chemistry, Cardiovascular Research Institute, University of California San Francisco, San Francisco, CA, USA. [4]Department of Biochemistry and Biophysics, University of California San Francisco, San Francisco, CA, USA. ✉e-mail: daniel.southworth@ucsf.edu

including Tauvid, a first-generation tau PET ligand that is FDA-approved and clinically available[22]. While second-generation PET tracers have been developed to reduce off-target binding and optimize pharmacokinetic properties[23,24], direct binding to disease-relevant tau filament folds is under characterized. Docking studies have predicted that PET tracers bind end-to-end with the plane of the aromatic rings parallel to the fibril axis[25,26], and a cryo-EM structure of the PET tracer APN-1607 at low resolution[27] has been modeled to have the same orientation. Conversely, the cryo-EM structure of AD tau PHFs incubated with the green tea molecule Epigallocatechin gallate (EGCG, a compound known to disaggregate amyloid filaments in vitro)[28] shows several unique densities along the filament surface, and model building indicated that the most well-defined density represented EGCG with aromatic rings stacked perpendicular to the fibril axis[29]. However, the molecular details of the interactions were not well resolved based on the density and multiple binding sites.

## Results

### GTP-1 binds specifically to a unique cleft in tau AD PHFs

Using cryo-EM, we sought to determine the co-structure of AD tau filaments and GTP-1 (Genentech Tau Probe 1), a high affinity (11 nM $K_d$), second-generation tau PET tracer that is currently in clinical trials (Fig. 1a)[30]. Tau filament samples purified from post-mortem frontal cortex tissue from a patient with AD (see Methods)[11] showed high infectivity in a cell-based assay[31] (Supplementary Fig. 1) and were incubated with 20 μM GTP-1 prior to vitrification. The micrograph images and 2D classification reveal well-resolved filaments primarily in the PHF conformation, with crossover distances between 700 and 800 Å (Supplementary Fig. 2). A minor population of straight filaments (SFs) was also identified; however, further structural characterization was not feasible due to limited abundance (Supplementary Fig. 2). Using standard helical reconstruction methods (Supplementary Table 1 and Methods), a structure of the PHF was determined with an overall resolution of 2.7 Å (Supplementary Fig. 3a, b) and is comprised of two protofilaments related by two-fold symmetry with a 2.37 Å rise and 179.45° twist (Supplementary Table 1), consistent with previously reported structures of PHFs prepared from AD brain tissue[11,13]. The central region surrounding the protofilament interface is at the highest resolution at ~2.5 Å and the periphery is at ~3.2 Å, indicating high resolution across the β-sheet core, as evidenced by well-resolved side chain densities (Supplementary Fig. 3c).

Remarkably, the structure reveals strong additional density that is indicative of the GTP-1 small molecule bound to a solvent-exposed cleft (Fig. 1b, c). This density appears identical in both protofilaments, indicating equivalent binding, considering two-fold symmetry was not enforced in the refinement. While other densities are present around the filament core, these are poorly resolved in comparison and similar to previously reported tau filament structures (Supplementary Fig. 4)[11,13]. Importantly, difference map analysis comparing the GTP-1 co-structure (tau PHF:GTP-1) to a previously determined PHF map (EMDB: 0259)[13] identifies that this density is uniquely present (Fig. 1d). Additional unresolved densities adjacent to the filament core are identical to those seen in previous PHF structures (Supplementary Fig. 4). These densities are unknown but may represent neuronal metabolites or additional disordered regions of tau outside the structured core, as previously reported[11]. The lack of additional sites of small molecule density in our structure establishes a specific, single-site interaction by GTP-1 and contrasts with other structural studies showing more heterogeneous small molecule binding to tau PHFs[25–27,29]. Notably, the 20 μM GTP-1 concentration used is well above the measured $IC_{50}$ (22 nM)[30] (see Methods), further supporting the specificity. The GTP-1 density reveals binding in a stacked, geometric repeat that precisely matches that of protein monomers in the fibril (Fig. 1e). This arrangement is distinct from previous studies reporting

binding end to end or parallel to the fibril axis[25–27] but is similar to the stacked EGCG-tau model[29].

An atomic model of the tau portion of PHF:GTP-1 was achieved by docking and refinement of the previous PHF structure solved in the absence of an exogenous ligand (Fig. 2a)[13]. The protofilaments form the canonical C-shaped cross-β fold found in AD that is comprised of the 3 R and 4 R tau domains (residues 306–378) and interact laterally via the antiparallel PGGGQ motif (residues 332–336). The overall filament structure is nearly identical to previous structures of AD PHFs (α-carbon RMSD = 0.5 Å) (Supplementary Fig. 5a). GTP-1 is bound in the cleft comprised of residues 351–360 (Fig. 2A), adjacent to the three-strand β-helix (β5–7) in the protofilament. Small differences are seen in the sidechains of the residues lining the binding pocket, namely Gln351, Lys353, Asp358, and Ile360 (Supplementary Fig. 5b).

Accurately modeling small molecule ligands is a notable challenge[32], and the tau PHF:GTP-1 structure presents additional difficulties due to the novel stacked arrangement of GTP-1 in which ligand-ligand interactions are likely making substantial contributions. Furthermore, while the tricyclic aromatic ring is rigid, the piperidine ring and fluoroethyl tail are highly flexible and difficult to model by standard methods (Supplementary Fig. 6). Our best modeling approach resulted from a combination of molecular mechanics to generate different conformers and density functional theory to perform constrained optimizations of dimers to capture small molecule–small molecule interactions, followed by refinement with Phenix[33]. The final modeled conformer yields excellent map-model agreement and is energetically reasonable (Fig. 2b, Supplementary Fig. 7, Supplementary Table 1, and Supplementary Dataset 1; see Methods). This map-model agreement, along with the fact that the ligand density has a similar resolution to the adjacent filament structure (~2.6 Å) and remains present at high sigma threshold values, indicates near-complete occupancy of GTP-1 (Supplementary Figs. 3c, 4b).

### GTP-1 pi-stacking complements small molecule–protein interactions

GTP-1 binds in the C-shaped groove of the PHF filament comprised of strands β6 and β7, which are separated by a kink at Gly355 that creates a concave cleft complementing the convex shape of the GTP-1 stack (Fig. 2b). We identify that each molecule of GTP-1 binds across three β-strands, making direct contacts with Gln351 in strand 1, Gln351 and Lys353 in strand 2, and Ile360 in strand 3, as well as the backbone between Gln351 and Lys353 in strands 1 and 2 (Fig. 2c). Notably, the piperidine ring and fluoroethyl tail of GTP-1 are parallel to the filament and project across two β-strands, appearing to contact the sidechain and backbone of Gln351 in both strands. Although the site is comprised of primarily polar residues, there is precise matching between apolar portions of the sidechains and those of the small molecule (Fig. 2d and Supplementary Fig. 8). The aliphatic carbon of Ile360 contacts C7 of the phenyl ring and the apolar carbons of the Gln353 sidechain line the section of the pocket occupied by the relatively nonpolar fluoroethyl tail. Specific hydrogen bonding interactions also make prominent contributions to the binding of GTP-1. Lys353 lies at the base of the concave binding groove, where it forms a bifurcated hydrogen bond with the benzimidazole nitrogen (2.8 Å N–N distance) and the pyrimido nitrogen (3.4 Å) of GTP-1, satisfying the hydrogen bonding potential of the buried polar atoms within the tricyclic aromatic ring. Lys353 also completes its hydrogen bonding potential by forming a strong salt bridge with Asp358 in the same strand and a weaker hydrogen bond with Asp358 in the adjacent strand. The oxygen of the Gln351 sidechain is well positioned to make a noncanonical hydrogen bond with the C–H bond of the beta carbon of the fluoroethyl tail, which points toward the fibril backbone. Indeed, when the protein-ligand interaction energy is broken down on a per residue basis, these

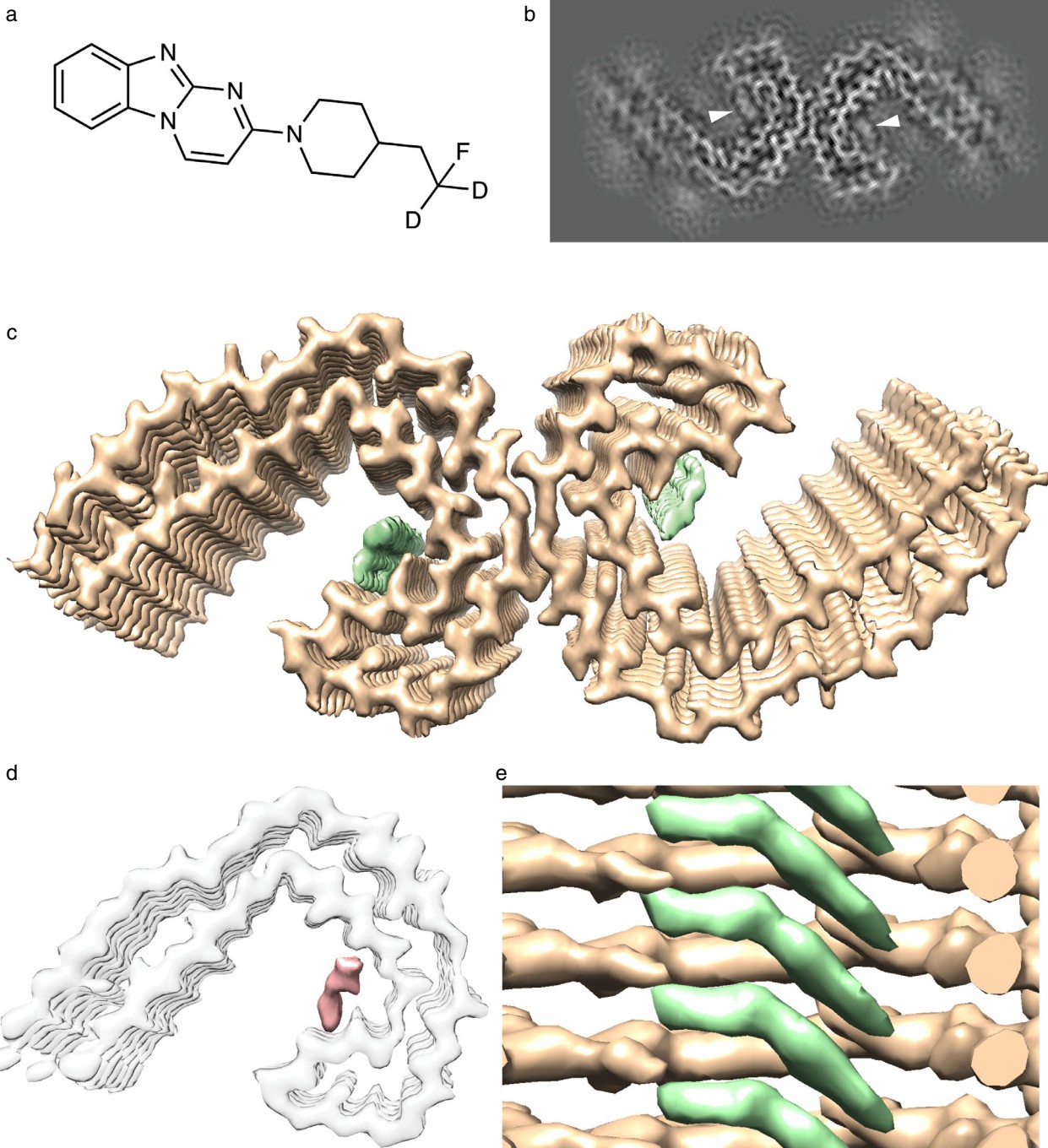

**Fig. 1 | Cryo-EM map of AD tau PHF with density for bound GTP-1. a** Chemical structure of GTP-1. **b** X-Y-slice view of the cryo-EM map of AD PHFs incubated with GTP-1. Extra density corresponding to GTP-1 is indicated by white triangles. **c** Cryo-EM map of tau PHF:GTP-1. The density corresponding to GTP-1 is colored in green. **d** Difference map (salmon density) between (C) and a previously determined apo- AD PHF map (EMDB: 0259), low-pass filtered to 3.5 Å. The density for the apo PHF protofilament (grey) is shown as a reference. **e** Side view of tau PHF:GTP-1 structure showing the ligand density (green) in a stacked arrangement with one molecule spanning across multiple rungs of the tau protofilament.

two residues contribute more than 50% of the overall interaction energy (Supplementary Table 2, note: these values do not account for desolvation effects). This tail orientation also allows for close van der Waals contacts with backbone atoms in two strands and for the interaction with the sidechain of Gln351 (Fig. 2e). Overall, there is remarkable physiochemical and geometric complementarity between GTP-1 and the binding cleft of the tau filament, which is unique to this cleft (Supplementary Fig. 8) and may explain the specificity of GTP-1 binding to this site.

Examining tau PHF:GTP-1, we observe that the GTP-1 heterocycles are situated at an optimal distance for pi-pi stacking (3.3–3.5 Å)[34], and GTP-1 forms an extended assembly scaffolded by the tau filament, reminiscent of supramolecular polymers that are highly cooperative[35]. Unlike those molecules, GTP-1 contains both a rigid heteroaromatic and flexible nonaromatic region (aromatic: pyrimido[1,2-a]benzimi-dazole; nonaromatic: 2-fluoro-4-ethylpiperidine) (Fig. 3a). To assess the favorability of these interactions, we performed Hartree–Fock London Dispersion calculations[36]. Each region of GTP-1 makes distinct

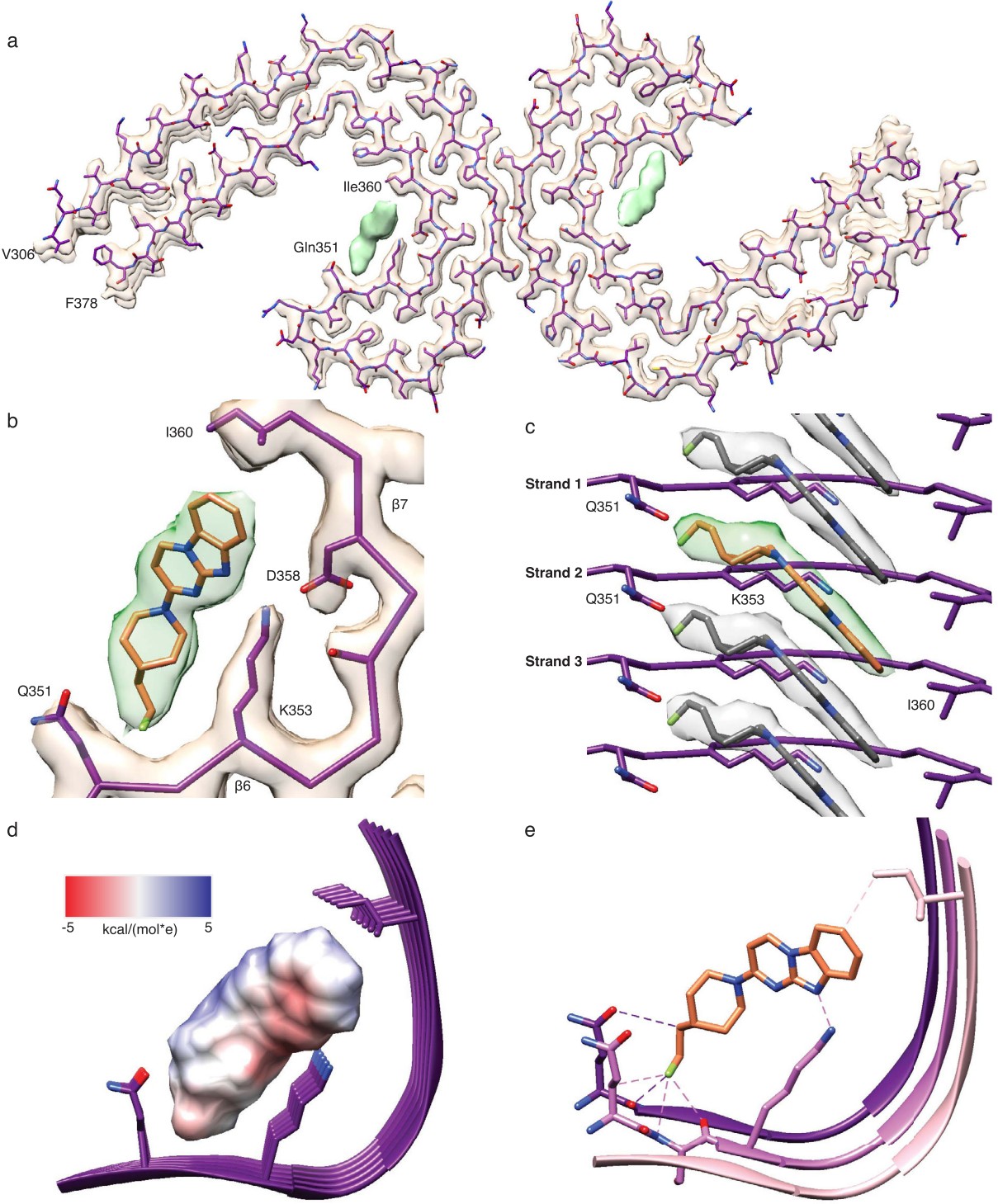

**Fig. 2 | Atomic model of tau PHF and bound GTP-1. a** Refined tau PHF atomic model fit into the PHF:GTP-1 density. **b** Map and model of the GTP-1 binding site with GTP-1 modeled into the density using a combination of molecule mechanics and density functional theory (DFT) approaches. **c** Side view of tau PHF:GTP-1 model, showing individual GTP-1 molecules fit at an angle relative to the backbone and making contact across 3 rungs of tau. **d** GTP-1 electrostatic (Coulomb) potential surface representation showing complementarity to the GTP-1 binding pocket. **e** Close contacts (<3.5 Å) of GTP-1 with sites in the binding pocket.

contributions to the overall interaction; the major component (16 kcal/mol, 57%) indeed originates from the aromatic-aromatic interaction, whereas the smallest contribution comes from the cross interaction of the nonaromatic region with the aromatic region (5 kcal/mol, 18%), and the remainder comes from the nonaromatic-nonaromatic interaction (7 kcal/mol, 25%) (Fig. 3b). Given that these subunits (aromatic and nonaromatic) have a similar surface area (340 Å² and 315 Å²,

respectively), this speaks to the electronic favorability of stacking aromatic molecules as opposed to nonaromatic molecules. Moreover, this analysis does not consider entropic and hydrophobic contributions, which will also favor more rigid, aromatic molecules. The "tilt" angle of GTP-1, which leads to each compound crossing multiple tau strands, is congruent to that formed by the z-axis of the fibril, which is defined by a 4.77 Å repeat in this amyloid filament (note: helical twist is

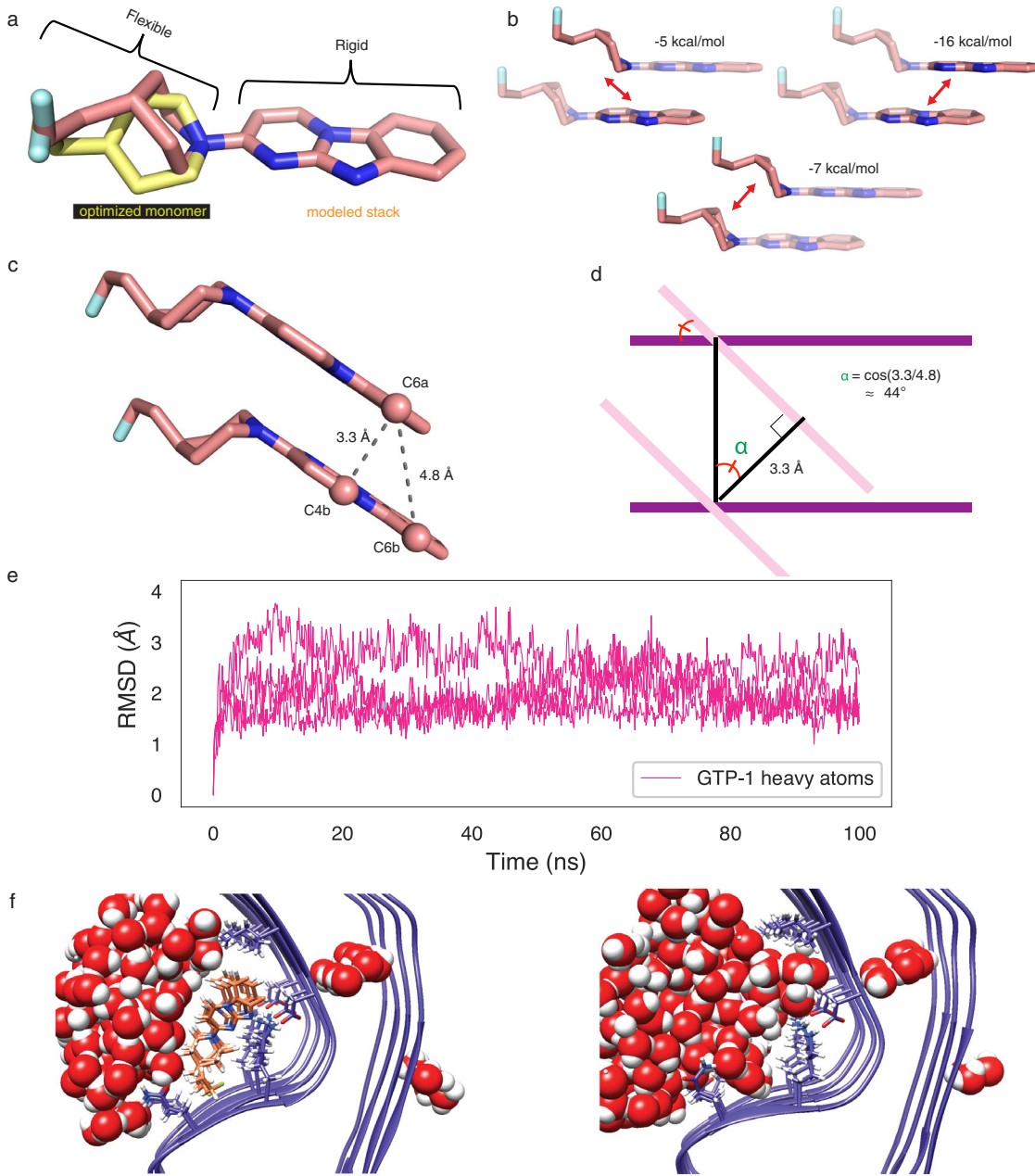

**Fig. 3 | Favorable ligand-ligand interactions support stacked arrangement.**
**a** Comparison of the structure of GTP-1 monomer from an unconstrained DFT optimization (yellow) with the final modeled structure optimized in the context of amyloid-imposed constraints (coral). **b** Energy decomposition of the GTP-1 stacking interaction in a dimer using an HFLD calculation. **c** Illustration of the stacked GTP-1 interactions demonstrating the slipped nature of the stack, the retention of the amyloid displacement vector, and the distance of the pi-pi interactions. **d** Abstracted depiction of how the crossing angle between the plane of the amyloid backbone and the plane of the heterocycle is determined by the amyloid displacement vector and the optimal dimer interaction distance. **e** The RMSD of the GTP-1 heavy atoms throughout the 100 ns MD simulation, showing the stability of the GTP-1 binding pose. **f** Representative final frames of a 100-ns MD simulation of tau PHF:GTP-1 (left) and unliganded tau PHF (right) demonstrate both the stability of the GTP-1 binding pose and the complete occlusion of water from the GTP-1 binding site throughout the trajectory.

negligible over a short assembly) and the distance between two aromatic rings along the normal to the plane, typically most favorable between 3.3 and 3.5 Å. This angle is then described by a simple cosine relationship between these two distances, here 44° (Fig. 3c, d). Given the commonality of these two constraints, we anticipate that the adoption of a tilted heterocycle relative to the amyloid backbone may prove to be a common motif for binding filament structures, as this allows for significant favorable pi-pi interactions between small molecules while maintaining the translational symmetry of the amyloid. Indeed, this arrangement was seen in the PET ligand flortaucipir bound to chronic traumatic encephalopathy (CTE) filaments[37].

## Discussion

The tau PHF:GTP-1 structure suggests a potentially powerful strategy for the discovery and design of small molecules that bind with high affinity to amyloids in both a sequence- and conformation-specific manner. Filaments present a unique challenge for small-molecule design because their accessible surfaces tend to be relatively flat. This limits the amount of surface area potentially lost upon binding of a monomeric small molecule, hence the propensity for docking studies to show face-on binding of flat small molecules to the amyloid[25,26]. However, the modeling of EGCG bound to one of the sites in tau PHFs by Eisenberg and colleagues indicates a similar ligand orientation to

our tau PHF:GTP-1 model, in which the rings lie perpendicular to and match the symmetry of the fibril[29]. Thus, this model, although at low resolution, suggests the potential generality of this motif.

These structures suggest that this polymeric motif may have favorable filament binding properties, and we performed several calculations in an attempt to further examine this potential favorability. Although GTP-1 forms multiple productive contacts with the amyloid, the surface area lost upon binding a single monomer is negligible at 0.3 Å². However, when two GTP-1 molecules stack, the overall loss of surface area increases to 85 Å² (most of which is the apolar face of GTP-1) and creates a large driving force associated with the burial of hydrophobic groups as additional monomers are added. That this effect is not observed when two monomers are separated by an unliganded binding site suggests the system may be cooperative (Supplementary Table 3). To further examine this cooperativity, we undertook single-point density functional theory (DFT) calculations for binding of one, two, and three molecules of GTP-1 to five strands of a truncated model (residues 351–360) of tau (Supplementary Fig. 9). Although the accuracy of the calculations is intrinsically limited due to their static nature and lack of explicit solvation, potential trends can be gleaned. Notably, the binding energy of a single tracer against the five strands is the same in all three potential binding sites. For two tracers bound in adjacent sites, the energy is the sum of the small molecule–protein binding energies and the small molecule–small molecule dimerization energy, suggesting positive cooperativity. The same trends continue with three tracers (the minimal model for an extended stack), suggesting the calculations are relevant to the overall assembly. In contrast, two tracers separated by an unliganded binding site (a minimal model for sparse binding) shows no favorable small molecule–small molecule binding energy (Supplementary Table 3). While the concentration of GTP-1 (20 μM) used here is much higher than the likely concentration of free CNS drugs typically found in the brain (~30 nM–3.5 μM)[38], cooperative systems such as this that bury hydrophobic surface area are predicted to preferentially form large assemblies even at low concentrations[35]. We also used molecular dynamics to simulate five stacked ligands centered in nine strands of both protofilaments and found the stacked assembly to be stable over 100 ns (Fig. 3e). Throughout the simulation, the GTP-1:tau and GTP-1:GTP-1 interactions seen in the experimental structure were maintained, and no penetration of water was observed into the dry protein–small molecule interface, confirming the geometric and electrostatic complementarity of stacked GTP-1 with this binding groove (Fig. 3f, Supplementary Dataset 2).

Moreover, the observed behavior, that both small molecule–protein and small molecule–small molecule interactions are local and that the latter are positively cooperative, is analogous to other well-studied biological systems. These systems, including the random coil-to-helix transition of a polypeptide or the binding of dye molecules to DNA, are well described by mathematical models[39–41]. This suggests a route forward to better understanding the thermodynamic and kinetic behavior of small molecule–amyloid interaction under physiological conditions. In addition to the tau PHF:EGCG and CTE-flortaucipir structures[29,37], templated assembly and symmetry matching have also been observed in the assemblies of similar aromatic molecules with globular proteins, although the limited size of the binding sites restricts the assembly size to a maximum of four molecules[42–44].

Rather than binding to a nondescript surface along a uniform β-sheet, the strong geometric and physical complementarity between GTP-1 and this unique cleft likely imparts considerable specificity for AD filaments (Fig. 4a). The local architecture of Gln351 to Ile360 that comprises the GTP-1 binding site is markedly different in filament structures of other tauopathies. In many cases, the key residues that form close contacts in the AD structure are either not solvent exposed or instead form a convex surface as opposed to the concave cleft

suitable for binding. Although CTE protofilaments have a similar C-shaped architecture to AD, this region of the CTE filament structure is defined by a much shallower angle formed by the kink at Gly355. This causes Ile360 to shift ~3 Å further from Gln351 than in the AD structure, which would result in the loss of the apolar interaction between Ile360 and C7 of the GTP-1 phenyl ring (Fig. 4b) that accounts for almost 20% of the ligand-protein interaction (Supplementary Table 2). Based on these structural differences, GTP-1 likely does not stack in this cleft of CTE filaments. While it is possible that GTP-1 binds to other β-sheet folds, it would likely involve an alternate mode of binding and different sequence elements within the tau filament structure.

The specific and stoichiometric binding of GTP-1 is distinct from established small molecule dyes, such as thioflavin T (ThT), that are known to heterogeneously bind many different types of amyloid folds[45,46]. In crystal structures of ThT bound to soluble proteins[47–49], it is observed to dimerize in a head-to-tail fashion. HFLD calculations demonstrate that the observed dimers have similar interaction energies as observed for GTP-1 suggesting that ThT can likely stack in a manner similar to GTP-1 (Supplementary Fig. 10). However, the geometric diversity of the observed structures likely due to the flatter nature of ThT is consistent with a variety of (stacked) binding modes being accessible. Moreover, in contrast to GTP-1 or flortaucipir, ThT lacks a strong hydrogen bond acceptor that could help localize the molecule to specific regions of the amyloid and seed stacking. As such, the ThT:filament stoichiometry is unknown, whereas we have shown that GTP-1 binds to AD PHF's in a 1:1 fashion. Thus, although GTP-1 does not possess intrinsic fluorescence, the development of fluorescent GTP-1 analogs with similar binding characteristics could serve as powerful biomarkers for the absolute quantification of PHFs in vivo or as biochemical tools for ex vivo experiments.

Symmetry matching, as observed in the structure of GTP-1 bound to PHFs from a patient with AD, may provide a powerful strategy to increase the druggability of available binding sites in filaments. In an emergent system such as this, small changes to the binding site likely confer a large effect on the binding of GTP-1. Thus, designing small-molecule compounds with high specificity and affinity for a single site within the amyloid filament conformation may be feasible. This analysis suggests that in the development of future tools for diagnostics and, potentially, therapeutics, an emphasis should be placed on heterocycles that stack favorably in the context of the amyloid axial symmetry and on achieving shape and electrostatic synergy with the targeted binding cleft. Understanding not only the amyloid assembly as a supramolecular entity but also the small molecule, reveals a previously unknown route to designing amyloid filament binding small molecules.

## Methods

### Ethical review process and informed consent

Alzheimer's disease tissue was obtained from the UCSF Neurodegenerative Disease Brain Bank (NDBB). These experiments were approved by the NDBB Institutional Review Board 11-05588 and the UCSF Memory and Aging Center Autopsy Program Institutional Review Board 12-10512. This research was conducted in accordance with the principles of the Declaration of Helsinki. Informed consent was obtained from the patient's next of kin, including for identifying data to be shared with researchers using tissue obtained from the UCSF NDBB.

### Experimental design

Samples were selected based on brain tissue availability and post-mortem neuropathological examination. Sex was not considered in this study, as the structure of AD filaments does not vary by sex. Randomization and blinding were not performed in this study, as there was only one patient sample, with the sample size limited by tissue availability.

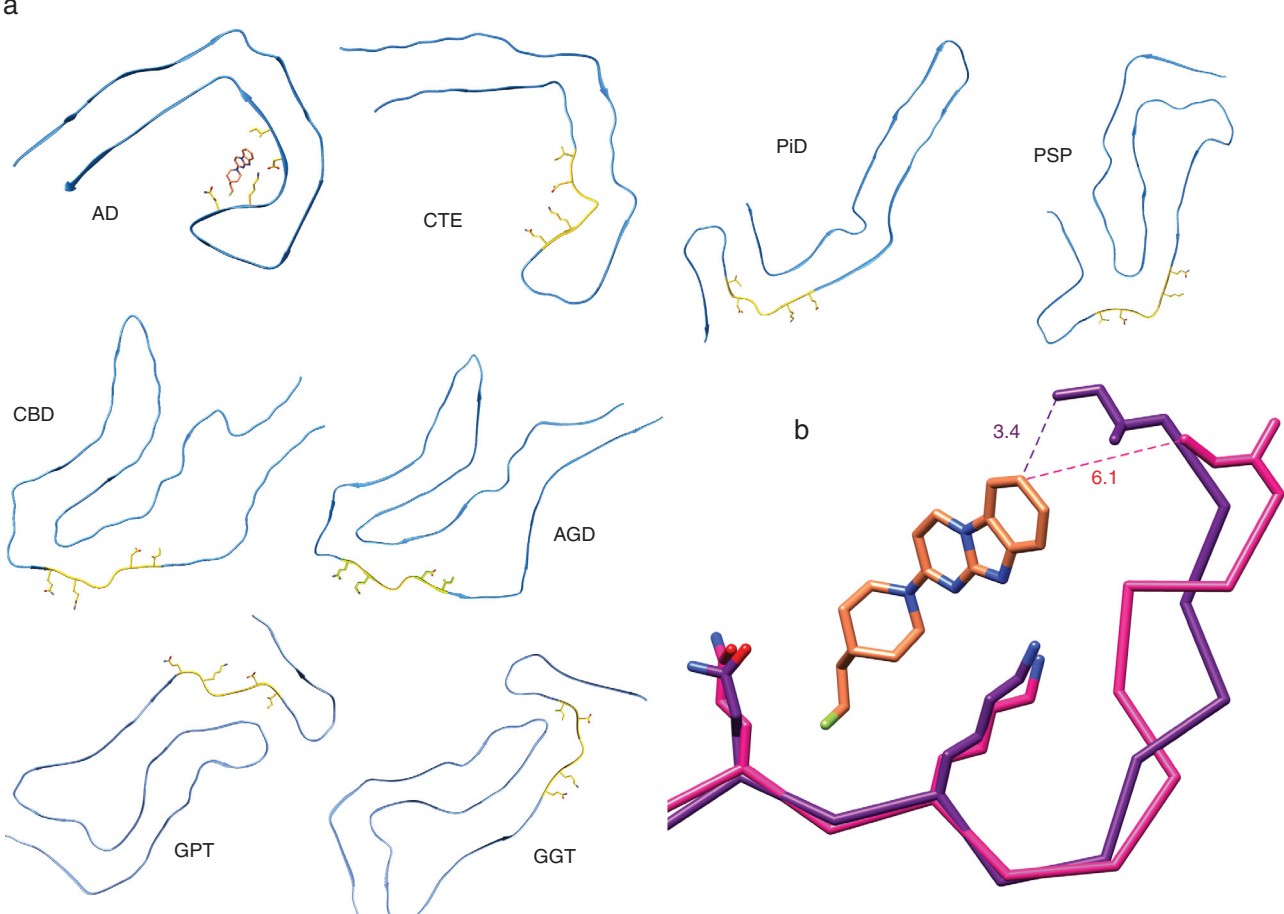

**Fig. 4 | Comparison of the GTP-1 binding pocket residues in other tau filament structures. a** The ligand binding pocket of GTP-1 is highlighted in gold, and the specific residues forming the binding pocket (Gln351, Lys353, Asp358, and Ile360) are shown. This binding pocket is unique to AD filaments compared to existing filament structures, thus indicating GTP-1 binding may be specific to the AD conformation. **b** Residues 351–360 in AD (purple) and CTE (pink) filament structures, with the rotamer of Lys353 matching that in our AD + GTP-1 structure. The change in concavity of the pocket would move Ile360 away from GTP-1 and would prevent a productive apolar interaction with C7 on the GTP-1 heterocycle.

## Purification of tau filaments

Filament purification was based on Fitzpatrick et al[11]. Briefly, 5 g of fresh-frozen frontal cortex tissue from an 88-year-old male patient with Alzheimer's disease was homogenized at 10 mL/g of tissue in 10 mM Tris-HCl (pH 7.4), 800 mM NaCl, 1 mM EGTA, and 10% sucrose. The homogenate was centrifuged at 20,000 $g$ for 10 minutes, and the supernatant was kept. The pellets were resuspended in 5 volumes of the same buffer, centrifuged again, and the 2 supernatants were combined. A final concentration of 1% N-laurosarcosinate (w/v) was added to the combined supernatant, and this mixture was incubated for 1 h at room temperature. It was then centrifuged at 100,000 $g$ for 1 h, and the pellets were resuspended in 30 volumes of 10 mM Tris-HCl (pH 7.4), 800 mM NaCl, 1 mM EGTA, 5 mM EDTA, and 10% sucrose. This was followed by another centrifugation at 20,100 $g$ for 30 minutes at 4 °C. The supernatant was kept and centrifuged at 100,000 $g$ for 1 h, and the final pellet resuspended in 20 mM Tris-HCl (pH 7.4) and 100 mM NaCl at a concentration of 10 µL/g frozen tissue.

## Negative stain imaging

Purified frontal cortex tissue was diluted 1:10 for a final concentration of 100 µL/g frozen tissue. 5 µL was added to a glow-discharged 400 mesh copper grid with a layer of amorphous carbon. After 30 seconds, the grid was blotted with filter paper, washed, and blotted twice with nanopore water. 5 µL of 0.75% uranyl formate was then added and blotted. Three more 5 µL aliquots of uranyl formate were added and

removed by vacuum aspiration. Images were collected on a Talos L120C (Thermo Fisher Scientific) operating at 120 kV and equipped with a Ceta-D (Thermo Fisher Scientific) camera.

## Tau quantification and infectivity

Total tau in the final purification fraction frozen on grids was quantified using a Total Tau cellular kit (HTRF, Cisbio). The output was read on a PHERAstar FSX plate reader (BMG LABTECH). A standard curve was generated using recombinant 0N4R tau.

Infectivity assays were then performed similarly to Woerman et al.[31], and an HEK293T cell line expressing the 4R repeat domain of tau (residues 243 to 375 in 2N4R tau) with mutations P301L and V337M fused to yellow fluorescent protein (YFP) at the C-terminus was used. Cells were cultured and plated in 1x Dulbecco's modified Eagle medium supplemented with 10% (vol/vol) fetal bovine serum. Cells were plated in a 96-well plate (3000 cells per well) with 0.1 µg/mL final concentration of Hoechst 33342. Cells were then returned to an incubator that maintained a humidified atmosphere of 5% $CO_2$ at 37 °C for 2 h. Samples were diluted to the appropriate tau concentration with Dulbecco's PBS (DPBS), mixed with Lipofectamine 2000 (Thermo Fisher Scientific; final concentration: 0.03%), and incubated for 1 h at room temperature. Samples were then added to the cells in six replicate wells and incubated for 3 days at 37 °C in a humidified atmosphere of 5% $CO_2$. Plates were imaged using the IN Cell Analyzer 6000 cell-imaging system (GE Healthcare). Images were then analyzed using the

IN Cell Developer software (GE Healthcare), with an algorithm that detects aggregated protein using pixel intensity and size thresholds in living cells. The output, DxA, is a measure of the size and brightness of these aggregates.

## Cryo-EM grid preparation and data collection

Purified frontal cortex tissue was incubated with 20 µM ligand for 45 minutes prior to freezing. Three µL of this mixture was added to a glow-discharged 200 mesh 1.2/1.3 R Au Quantifoil grid for 10 seconds before blotting for 2 seconds. A second 3 µL aliquot was added for 3 seconds and blotted for 1 second before being plunge frozen in liquid ethane using a FEI Vitrobot Mark IV (Thermo Fisher Scientific). Super-resolution movies were collected at a nominal magnification of X105,000 (physical pixel size: 0.417 Å per pixel) on a Titan Krios (Thermo Fisher Scientific) operated at 300 kV and equipped with a K3 direct electron detector and BioQuantum energy filter (Gatan, Inc.) set to a slit width of 20 eV. A defocus range of 0.8 to 1.8 µm was used with a total exposure time of 2.024 seconds fractionated into 0.025-second subframes. Movies were motion-corrected using MotionCor2[50] in Scipion[51] and were Fourier cropped by a factor of 2 to a final pixel size of 0.834 Å per pixel.

## Image processing

For GTP-1, 15,160 micrographs were collected, and all processing was done in RELION 3.1[52]. Dose-weighted summed micrographs were imported into RELION 3.1. The contrast transfer function was estimated using CTFFIND-4.1. Filaments were manually picked and then segments were extracted with a box size of 900 pixels downscaled to 300 pixels. A larger box size of 1200 pixels downscaled to 300 pixels was used to estimate the filament crossover distance. Contaminants and segments contributing to straight filaments were separated out using reference-free 2D class averaging. The remaining segments were re-extracted with a box size of 288 pixels without downscaling. The map from EMDB 0259[13] low-pass filtered to 15 Å was used as an initial model. One or more rounds of 3D classification with image alignment were performed, with helical rise and tilt parameters fixed to eliminate obvious junk particles. Local rise and tilt were fixed during a first round of 3D auto-refinement using $C_1$ symmetry and a PHF map low-pass filtered to 10 Å. A second round of 3D auto-refinement was run imposing $C_{21}$ symmetry and allowing rise and twist parameters to vary, using the map from the first auto-refinement low-pass filtered to 4.5 Å as a model. Contrast transfer function (CTF) refinement was then run, fitting the defocus and astigmatism, as well as estimating $4^{th}$ order aberrations. These particles were then used in a 3D classification job allowing the rise and twist to vary but without image alignment. Particles contributing to the highest resolution map(s) were selected, and a final 3D auto-refinement was run. Maps were sharpened using the standard post-processing procedures in RELION. Full statistics are shown in Supplementary Table 1.

A reconstruction of straight filaments was attempted using the same workflow, but a high-resolution structure was unable to be obtained, even after extensive 2D and 3D classification.

## Refinement of Tau PHF

Prior to ligand placement, a single strand of a previously solved PHF model (PDB: 6HRE)[13] [https://doi.org/10.2210/pdb6HRE/pdb] was refined against the density using Phenix[33]. Refinement of side chains in the GTP-1 binding pocket was done in COOT[53]. This apo model was then translated to give a stack spanning five rungs and validated in Phenix.

## Computation and modeling of GTP-1

**General Considerations.** All molecular mechanics-based conformer searches were performed using the ConfGen tool in Maestro[54]. The OPLS4 forcefield[55] was used, and an energy threshold of 21 kJ/mol

(or 5.02 kcal/mol) was used. All DFT calculations were performed using ORCA 5.0.3[56]. Optimizations were performed using the BP86 functional[57,58] and the def2-SVP basis sets[59] with an auxiliary basis set approximation[60], a dispersion correction[61], and a solvent polarization model (CPCM)[62]. Dichloromethane was used as the solvent model because it has dielectric properties similar to those found in proteins[63]. In cases of unconstrained optimization, a numerical frequency calculation was performed to confirm that the geometry was at a global minimum. In cases where a constrained optimization was performed, the electronic energy was used for comparison as an estimate of the enthalpy, which is valid assuming a similar zero-point energy, vibrational energy, rotational energy, and translational energy. This estimate is necessary because these systems are not at a global minimum, and, thus, the exact calculation of the enthalpy and entropy via DFT will be prone to errors. Given that systems of similar size are being compared, the following estimate should be valid for determining relative energies:

$$U = E_{electronic} + E_{zero\,point\,energy} + E_{vibrational} + E_{rotational} + E_{translational} \quad (1)$$

$$H = U + k_B T \quad (2)$$

**Modeling of GTP-1.** A DFT-minimized monomer of GTP-1 was used as the input for the initial conformer search (0.5 Å RMSD) in Maestro. Outputs (43) were clustered by the position of the piperidine ring (maximum atom distance <0.5 Å) ignoring the fluoroethyl tail. The centroids and their fit to the density can be seen in Supplementary Fig. 7c. From the best-fit conformer, a dimer was then generated taking into account the translational vector of the amyloid (the rotational element is considered to be negligible over two units). The dimer was then optimized for a series of torsional angles and the electronic energies were compared (Supplementary Fig. 7e). The lowest energy torsion also improved the fit to the density, so that was used for a further conformational search in Maestro. In this search, all of the atoms of the tricyclic aromatic and the piperidine ring were held constant (i.e., only the fluoroethyl tail was varied) and the outputs were required to have at least one atom that varied by more than 0.1 Å. Both small molecule–small molecule and small molecule–protein clashes were then considered for the outputs (13). Clashes were defined as two heavy atoms (C, N, O, F) with a distance of <2.5 Å. All outputs that passed the clash filter (5) were again subjected to a constrained DFT optimization, and that final output was compared to the cryo-EM density. Selecting the best output based on the density, a final refinement was done in Phenix. See Supplementary Dataset 1.

For the modeled conformer, the interaction energy can be evaluated via a Hartree−Fock London Dispersion calculation, which decomposes the overall energy of a system into the energy of the individual units and the energy arising from their interactions. This calculation was performed using the def2-TZVP(-f) basis set[59] with the auxiliary basis sets def2/J[60] and def2-TZVP/C[64] in a continuous polarized solvent model[60]. The interaction energy of −26 kcal/mol for a GTP-1 dimer can be decomposed into the components coming from the aromatic and nonaromatic subregions by performing calculations on those individual pairs with a proton capping the portion of the molecule that was removed.

**Binding to the amyloid.** Estimates of the interaction energy of the small molecule(s) with the protein could be readily achieved via single point calculations in the apo- and holo-state. To speed calculations, a truncated active site region was considered, consisting of residues 351–360 of a given strand with protons added to cap the backbone. Five strands in total were considered. A single PET tracer appears to interact with three strands of the amyloid backbone via visual

inspection, so there were three binding sites across the five strands. Single point calculations were performed with a single GTP-1 in each of the binding sites, and the energies were confirmed to be constant, suggesting that protein–small molecule interactions are local (Supplementary Table 1). If long-range interactions were observed, then positioning GTP-1 in the middle binding site should be more favorable. This validates the model size.

We also performed calculations in which GTP-1 occupies both the top two or both the bottom two binding sites, which were isoenergetic. However, calculations with two GTP-1 spaced out (i.e., the middle binding site is empty) show lower energy. A final calculation in which all three sites are occupied by GTP-1 confirms the trends shown with the two sites. As every additional GTP-1 added after three effectively introduces another unit into the interior of the stack, a calculation on a larger system is not needed.

The $\Delta\Delta E_{binding}$ term (Supplementary Table 1) was evaluated by comparing the energy to the energy of binding one GTP-1 in the middle of the stack adjusted for the stoichiometry. The negative terms seen for the 2 GTP-1 (top), 2 GTP-1 (bottom), and 3 GTP-1 are a result of the cooperative effect of the GTP-1 interactions. The magnitude of that interaction (about −19 kcal/mol) suggests that DFT probably slightly underestimates this dispersion-based interaction, a well-known phenomenon[65].

The surface area was evaluated using the same truncated systems with the get area feature in PyMOL (solvent turned on and the dot density set to the maximum)[66].

**Atomistic molecular dynamics simulations of ligand-bound tau**
**Simulation parameters and analysis.** The MD system was prepared using AmberTools in Amber18[67,68]. The N-termini of the cryo-EM structure were acetylated, while the C-termini were amidated. The electrostatic potential of GTP-1 was calculated using Gaussian09[69], which was subsequently used to fit partial charges of the molecule. Additional GTP-1 parameters were generated using AmberTools. The structure was then solvated with SPC/E-modeled waters in an octahedron with 8 Å buffer from the protein, and the system was neutralized by adding Cl- ions. All simulations were performed using Amber18 with the ff14SB forcefield[70,71]. Simulations began with 1000 restrained steepest-descent minimization steps before switching to a maximum of 5000 steps in conjugate gradient steps. The system was then heated up to 300 K over 50 ps in NVT equilibration with Langevin thermostat control of temperature and harmonic restraints on protein and small molecule atoms with a 10 kcal/(mol·Å$^2$) force constant. The system was then switched to NPT, which used the Monte Carlo barostat to maintain pressure at 1 atm. The restraints were gradually removed over 1 ns, and the simulation progressed to an unrestrained production run for 100 ns.

The systems were simulated under periodic boundary conditions, employing the SHAKE algorithm with 2.3 fs timesteps. Particle Mesh Ewald was used for long-range electrostatics, and non-bonded interactions were cut off at 8 Å. Two independent simulations of GTP-1–bound paired helical filaments were performed, for a total of four simulated protofilaments. Using MDAnalysis[72,73], time series of RMSDs were calculated to the starting cryo-EM structure as a measure of conformational stability. See Supplementary Dataset 2.

**Reporting summary**
Further information on research design is available in the Nature Portfolio Reporting Summary linked to this article.

## Data availability
Cryo-EM maps and atomic coordinates have been deposited in the EMDB and PDB with accession codes: EMD-29458 and PDB 8FUG. Any other relevant data are available from the corresponding author upon request.

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

## Acknowledgements

We thank Bill Seeley and the UCSF Neurodegenerative Disease Brain Bank for providing patient tissue for this study. We also thank Neil Vasdev and MedChem Imaging for providing GTP-1 PET ligand. This work was funded by the Rainwater Charitable Foundation (D.R.S.), and the National Institutes of Health (P01AG002132: D.R.S., S.B.P., and W.F.D.; F32GM139379: M.J.C.). The UCSF Neurodegenerative Disease Brain Bank receives funding support from NIH grants P30AG062422, P01AG019724, U01AG057195, and U19AG063911, as well as the Rainwater Charitable Foundation and the Bluefield Project to Cure FTD.

## Author contributions

G.E.M. purified patient tissue samples, prepared EM grids, collected data, performed cryo-EM image analysis, performed model building and refinement, developed figures, and wrote and edited the manuscript. M.J.C. performed calculations, performed model building and refinement, developed figures, and wrote and edited the manuscript. S.T. performed calculations and developed figures. E.T. operated the Krios microscope and helped with data collection. J.L. performed tau quantification and infectivity assays. S.B.P. supervised the project and edited the manuscript. N.A.P. provided critical ideas and methods and edited the manuscript. W.F.D. supervised the project and edited the manuscript. D.R.S. initiated the project, supervised the project, and edited the manuscript.

## Competing interests

S.B.P. is the founder of Prio-Pharma, which did not contribute support for this study. W.F.D. is a member of the scientific advisory boards of Alzheon Inc., Pliant, Longevity, CyteGen, Amai, and ADRx Inc., none of which contributed support for this study. The remaining authors declare no competing interests.
