## [Peer Review File · Nature Communications]

REVIEWERS' COMMENTS

Reviewer #1 (Remarks to the Author):

Merz et al. applied cryo-EM to determine the binding site of the PET ligand GTP-1 bound to Alzheimer's disease patient-derived tau paired-helical filaments. Based on the electron density, the repeating position of GTP-1 was modeled using molecular docking, MD simulations and DFT calculations.

The novel aspect of the work is that cryo-EM allowed to reveal the position of the ligand bound to the brain-extracted tau fibrils (resolution 2.7 Ang.) However, the fibril structure without ligand has been known before. Moreover, the authors were not the first to achieve this, yet the other work (DOI:10.1007/s00401-021-02294-3, 10.1038/s41467-022-32951-4) studied other tau fibril - ligand systems.

Modeling of the ligand in the binding site was done very rigorously. However, it is rather straightforward to come up with sensible docking poses if the docking site is as well defined as in the current case thanks to the high-resolution electron density. The novel aspect is that the GTP-1 docking pose was repeated (stacked) along the fibril aspect and the orientation between the GTP-1 molecules optimized for favorable ligand-ligand interactions. The much discussed stabilization of this arrangement due to pi-pi aromatic stacking between the GTP-1 molecules is not surprising though. What would have been more, or at least equally interesting is an analysis why GTP-1 prefers to bind at the indicated site and not anywhere else on the fibril, i.e., what other stabilizing factors are there? Fig. 2 is not sufficient for this; a proper energy decomposition revealing the interactions between GTP-1 and the tau residues is needed.

Apart from this, a question is how relevant the ligand repetition along the fibril axis is in a biological context. In the current experiments, the PET ligand was added at a concentration that allows stoichiometric binding between the fibrils and GTP-1. However, can such high GTP-1 concentrations be reached in brains? If not, of what relevance is the much discussed stacking between GTP-1 molecules then?

Finally, the authors discuss that the binding of GTP-1 to the C-shaped cavity present in AD fibrils "likely imparts considerable specificity for AD filaments" (Fig. 4). The word "likely" already indicates that this is a hypothesis. The authors should make some effort that this is more than a speculation by testing the binding of GTP-1 to tau inclusions derived from other diseases.

In summary, the manuscript is based on solid work that includes some novel aspects. However, similar work has already been published, which limits the novelty of the current study. In addition, further analysis should be conducted (as explained above) to support the observations made by the authors.

Reviewer #2 (Remarks to the Author):

Merz and coauthors provide a cryo-EM structure of a complex between AD tau fibrils and GTP-1, a second-generation PET ligand. In addition, they present extensive computational modeling of the key ligand-tau and ligand-ligand interactions. The findings are novel, well-presented, and illuminating with respect to the important task of developing more specific diagnostic imaging probes for AD and other amyloid diseases. The work also nicely complements previous analyses of ECGC binding by the Eisenberg group. I have only a few minor suggestions for improvement.

- 1) Ext. data Fig. 4: The legend mentions arrows, but I don't see any.
- 2) Extended Data Fig. 6: I would find it easier to compare the models in a and b if comparable parts of the GTP-1 molecule (notably the piperidine ring) were given the same color.
- 3) L138: insert "toward"?

Reviewer #3 (Remarks to the Author):

The manuscript entitled "Stacked Binding of a PET ligand to Alzheimer's tau paired helical filaments" submitted to Nature Communications (NCOMMS-22-53319-T) by Southworth and his colleagues at UCSF is an excellent paper that should be published without delay after minor revisions. I do not need to see this paper again, because if it was published as it is now, this would be a substantial contribution to the literature that enables others and thus will be highly cited.

Arguably the most important conclusion from the authors paper is not mentioned, provided I understand their data correctly. They have established a 1:1 stoichiometric relationship between tau protomers in amyloid and GTP-1 binding—something the field has needed for doing proper kinetic measurements. Use Thioflavin T fluorescence to measure amyloid aggregation is far from ideal because the binding stoichiometry of Thioflavin T to amyloidogenic protomers is not established to the best of my knowledge (more than my 30 min survey of the literature is required to confirm this). Exactly how thioflavin T binds and its stoichiometry relative to amyloidogenic protomers seems to be unknown, thus any kinetic expert would conclude that thioflavin's utility to follow aggregation kinetics is very limited. While authors state that the GTP-1 binding model is similar to the EGCC-tau model, which isn't as well resolved or as relevant, they don't say anything about thioflavin T structures or lack thereof, and the similarities with their GTP-1 amyloid structures. Comparisons to thioflavin amyloid binding would improve their already excellent paper. I can't find anything in the literature about the fluorescence of GTP-1 bound to tau NFT, if this bound structure is fluorescent (GTP-1 looks like a fluorophore to me, it has an auxochromic amine), the community should shift to the GTP-1 fluorophore for measuring NFT formation / amyloid assembly kinetics. If not somebody should make a fluorescent GTP-1 analog owing to its established 1:1 GTP-1 amyloid protomer binding stoichiometry—seems like a discussion along these lines would add a lot to their Nature Comm. Paper.

In the abstract, the statement that "Tau filaments adopt disease-specific conformations that self-propagate and drive neuronal loss....." is almost certainly an overstatement given what we know. While the tafamidis, tofersen, and lecanemab clinical trial results clearly demonstrate that protein aggregation drives the progression of amyloid diseases, a structure proteotoxicity relationship has not yet been established for any amyloid disease. Moreover, the rate of amyloid clearance in the tafamidis and tofersen data is much slower than the rate of the clinical response, thus amyloid may not be the main non-native conformation that is driving the disease. I simply suggest integrating "putatively" into this statement: "Tau filaments adopt disease-specific conformations that self-propagate and putatively drive neuronal loss....." in a revised version of the paper.

Are the non-GTP-1 densities around the amyloid fibril core lipid or carbohydrate, addressable by mild organic solvent extraction and enzyme treatment, respectively, likely beyond the scope of this paper but it seems important to know this.....

We thank the reviewers for taking the time to provide thoughtful feedback. Overall, comments were positive, indicating: “the findings are novel, well-presented, and illuminating with respect to the important task of developing more specific diagnostic imaging probes for AD and other amyloid diseases”, “published as it is now, this would be a substantial contribution to the literature”, and that “modeling of the ligand in the binding site was done very rigorously.” The reviewers also asked insightful questions and offered helpful suggestions, which we address here. Based on our structure, we hypothesize that other molecules with significant pi-character and which can adopt relatively flat conformations will stack similarly along the fibril axis. In support of our findings, another known PET tracer with these characteristics, flortaucipir, has recently been shown (Shi. et al (DOI: 10.1101/2022.12.15.520545)) to bind in a similarly stacked arrangement but with reduced interactions with the filament core compared to our structure. Based on helpful reviewer suggestions, we have done single point calculations on conformations of the established amyloid binder ThT found in complexes with soluble proteins, and we conclude that this compound indeed exhibits favorable stacking properties. However, it is known to bind many amyloid conformations, and thus interacts nonspecifically (DOI: 10.1038/s41598-017-12864-9). We postulate that while general amyloid binding molecules like ThT may also stack along the fibril, other interactions, such as the presence of strong hydrogen bond acceptors such as the benzimidazole and pyrimido nitrogens on GTP-1, are likely necessary for interaction at specific sites. We have addressed the reviewer comments and concerns below and include references to specific changes to text and figures:

Reviewer #1 (Remarks to the Author):

The novel aspect of the work is that cryo-EM allowed to reveal the position of the ligand bound to the brain-extracted tau fibrils (resolution 2.7 Ang.) However, the fibril structure without ligand has been known before. Moreover, the authors were not the first to achieve this, yet the other work (DOI:10.1007/s00401-021-02294-3, 10.1038/s41467-022-32951-4) studied other tau fibril - ligand systems.

We agree that the AD conformation has been well described previously. However, the GTP-1 PHF structure is the first specific small molecule bound to an AD conformation at high-resolution. The structures mentioned by the reviewer both show density in multiple binding sites, with that density being too low-resolution to unambiguously model the conformation of the small molecule.

What would have been more, or at least equally interesting is an analysis why GTP-1 prefers to bind at the indicated site and not anywhere else on the fibril, i.e., what other stabilizing factors are there? Fig. 2 is not sufficient for this; a proper energy decomposition revealing the interactions between GTP-1 and the tau residues is needed.

We have taken this reviewer’s insightful suggestion and performed an energy decomposition of the interactions of GTP-1 with the tau fibril (see Supplementary Table 2, now mentioned on lines 146-147 of the main text). We have adopted the same methodology that we used for calculating the small molecule-small molecule interaction energy (Hartree-Fock London Dispersion calculation). Calculation of the energy derived from interactions with the entire binding pocket (defined as three strands consisting of residues numbered 351-360) is ~16 kcal/mol of monomer.

On a per GTP-1 basis this is similar to that for the small-molecule-small molecule interaction (13 kcal/mol of monomer).

In the current experiments, the PET ligand was added at a concentration that allows stoichiometric binding between the fibrils and GTP-1. However, can such high GTP-1 concentrations be reached in brains? If not, of what relevance is the much discussed stacking between GTP-1 molecules then?

We agree that concentrations of GTP-1 *in vivo* will be lower than our concentration during vitrification, as the concentrations of CNS drugs in the brain are typically ~30 nM – 3.5 μ M (DOI: 10.1007/s11095-007-9502-2). However, given the favorability of the stacking we determined through HFLD calculations, we believe that stacking would still occur at these concentrations. Previously, a K_d of binding for GTP-1 to tau was measured in frozen, AD-patient brain tissue as 10.8 nM (DOI: 10.1007/s00259-019-04399-0). As discussed in this publication on supramolecular polymers (DOI: 10.1021/jacs.0c13081), the measured K_d likely represents the K_d for the transition from the monomer to fully oligomerized state. The burial of hydrophobic surface will likely drive these types of oligomers to form large assemblies even at low concentrations.

Finally, the authors discuss that the binding of GTP-1 to the C-shaped cavity present in AD fibrils “likely imparts considerable specificity for AD filaments” (Fig. 4). The word “likely” already indicates that this is a hypothesis. The authors should make some effort that this is more than a speculation by testing the binding of GTP-1 to tau inclusions derived from other diseases.

We agree with the reviewer that it would indeed be useful to know whether GTP-1 binds to tau filaments from other diseases. Approaches to address this would require obtaining and purifying tau filaments from other patient-derived sources (such as PSP or CBD, which are not trivial to acquire) and potentially solving numerous additional high-resolution cryo-EM structures. As this is a significant undertaking, and with limited tissue available, we consider this a future direction and outside the scope of the current manuscript. Additionally, we have found that there are no substantial fluorescence or absorbance properties for GTP-1, thus binding assays would be challenging, requiring radiolabeling or modification of the ligand.

Reviewer #2 (Remarks to the Author):

Merz and coauthors provide a cryo-EM structure of a complex between AD tau fibrils and GTP-1, a second-generation PET ligand. In addition, they present extensive computational modeling of the key ligand-tau and ligand-ligand interactions. The findings are novel, well-presented, and illuminating with respect to the important task of developing more specific diagnostic imaging probes for AD and other amyloid diseases. The work also nicely complements previous analyses of ECGC binding by the Eisenberg group. I have only a few minor suggestions for improvement.

We thank this reviewer for their kind words. All of the changes suggested below have been made.

1) Ext. data Fig. 4: The legend mentions arrows, but I don't see any.

Arrows have been added to Supplementary Figure 4.

2) Extended Data Fig. 6: I would find it easier to compare the models in a and b if comparable parts of the GTP-1 molecule (notably the piperidine ring) were given the same color.

Supplementary Figure 6 has been edited so that both panels now show GTP-1 in grey.

3) L138: insert "toward"?

"Toward" has been inserted.

Reviewer #3 (Remarks to the Author):

The manuscript entitled "Stacked Binding of a PET ligand to Alzheimer's tau paired helical filaments" submitted to Nature Communications (NCOMMS-22-53319-T) by Southworth and his colleagues at UCSF is an excellent paper that should be published without delay after minor revisions. I do not need to see this paper again, because if it was published as it is now, this would be a substantial contribution to the literature that enables others and thus will be highly cited.

We thank this reviewer for their kind words.

Comparisons to thioflavin amyloid binding would improve their already excellent paper. I can't find anything in the literature about the fluorescence of GTP-1 bound to tau NFT, if this bound structure is fluorescent (GTP-1 looks like a fluorophore to me, it has an auxochromic amine), the community should shift to the GTP-1 fluorophore for measuring NFT formation / amyloid assembly kinetics. If not somebody should make a fluorescent GTP-1 analog owing to its established 1:1 GTP-1 amyloid protomer binding stoichiometry-seems like a discussion along these lines would add a lot to their Nature Comm. Paper.

The reviewer is correct, that the binding mode of ThT to amyloid filaments is unknown. It has been shown that ThT binds to filaments in a heterogeneous fashion (DOI: 10.1038/s41598-017-12864-9), likely making high-resolution structural determination of a ThT-amyloid complex challenging. Further, crystal structures show a head-to-tail orientation of ThT dimers in complex with soluble proteins (PDBs: 3MYZ, 6ESY, and 6C6W). Calculations suggest that these dimerization modes are favorable (we have added Supplementary Fig. 10 showing this), and head-to-head dimers (or oligomers) that could bind amyloids are likely possible. However, unlike GTP-1 or flortaucipir, ThT lacks a strong hydrogen bond acceptor that could help localize the molecule to specific regions of the amyloid and seed stacking. Both the numerous pathways for assembly and the lack of a localization method could account for the heterogeneous binding and result in variable stoichiometry. Thus, GTP-1 more suitable candidate for quantification of PHFs.

We tested the potential fluorescence of GTP-1 upon binding to filaments, and found that it did not fluoresce. However, a fluorescent appendage could be attached to positions on the piperidine or tricyclic aromatic rings which face solvent, as these would likely still retain their ability to stack and specificity for PHFs.

A paragraph discussing GTP-1 and ThT has been added to the manuscript (lines 240-253).

In the abstract, the statement that “Tau filaments adopt disease-specific conformations that self-propagate and drive neuronal loss.....” is almost certainly an overstatement given what we know. While the tafamidis, tofersen, and lecanemab clinical trial results clearly demonstrate that protein aggregation drives the progression of amyloid diseases, a structure proteotoxicity relationship has not yet been established for any amyloid disease. Moreover, the rate of amyloid clearance in the tafamidis and tofersen data is much slower than the rate of the clinical response, thus amyloid may not be the main non-native conformation that is driving the disease. I simply suggest integrating “putatively” into this statement: “Tau filaments adopt disease-specific conformations that self-propagate and putatively drive neuronal loss.....” in a revised version of the paper.

We have revised the abstract, and in doing so, have softened our statement on neuronal loss. It now reads “...filaments adopt disease-specific cross- β amyloid conformations that self-propagate and are implicated in neuronal loss.”

Are the non-GTP-1 densities around the amyloid fibril core lipid or carbohydrate, addressable by mild organic solvent extraction and enzyme treatment, respectively, likely beyond the scope of this paper but it seems important to know this.....

We agree with the reviewer that this is an important open question in the field. We have added a statement in the manuscript about the potential nature of these densities (lines 93-96).